# Learning Gaussian Graphical Models with Observed or Latent FVSs

**Ying Liu**
Department of EECS
Massachusetts Institute of Technology
liu_ying@mit.edu

**Alan S. Willsky**
Department of EECS
Massachusetts Institute of Technology
willsky@mit.edu

## Abstract

Gaussian Graphical Models (GGMs) or Gauss Markov random fields are widely used in many applications, and the trade-off between the modeling capacity and the efficiency of learning and inference has been an important research problem. In this paper, we study the family of GGMs with small feedback vertex sets (FVSs), where an FVS is a set of nodes whose removal breaks all the cycles. Exact inference such as computing the marginal distributions and the partition function has complexity $\mathcal{O}(k^2 n)$ using message-passing algorithms, where $k$ is the size of the FVS, and $n$ is the total number of nodes. We propose efficient structure learning algorithms for two cases: 1) All nodes are observed, which is useful in modeling social or flight networks where the FVS nodes often correspond to a small number of highly influential nodes, or hubs, while the rest of the networks is modeled by a tree. Regardless of the maximum degree, without knowing the full graph structure, we can *exactly* compute the maximum likelihood estimate with complexity $\mathcal{O}(kn^2 + n^2 \log n)$ if the FVS is known or in polynomial time if the FVS is unknown but has bounded size. 2) The FVS nodes are latent variables, where structure learning is equivalent to decomposing an inverse covariance matrix (exactly or approximately) into the sum of a tree-structured matrix and a low-rank matrix. By incorporating efficient inference into the learning steps, we can obtain a learning algorithm using alternating low-rank corrections with complexity $\mathcal{O}(kn^2 + n^2 \log n)$ per iteration. We perform experiments using both synthetic data as well as real data of flight delays to demonstrate the modeling capacity with FVSs of various sizes.

## 1 Introduction

In undirected graphical models or Markov random fields, each node represents a random variable while the set of edges specifies the conditional independencies of the underlying distribution. When the random variables are jointly Gaussian, the models are called *Gaussian graphical models* (GGMs) or *Gauss Markov random fields*. GGMs, such as linear state space models, Bayesian linear regression models, and thin-membrane/thin-plate models, have been widely used in communication, image processing, medical diagnostics, and gene regulatory networks. In general, a larger family of graphs represent a larger collection of distributions and thus can better approximate arbitrary empirical distributions. However, many graphs lead to computationally expensive inference and learning algorithms. Hence, it is important to study the trade-off between modeling capacity and efficiency.

Both inference and learning are efficient for tree-structured graphs (graphs without cycles): inference can be computed exactly in linear time (with respect to the size of the graph) using belief propagation (BP) [1] while the learning problem can be solved exactly in quadratic time using the Chow-Liu algorithm [2]. Since trees have limited modeling capacity, many models beyond trees have been proposed [3, 4, 5, 6]. Thin junction trees (graphs with low tree-width) are extensions of trees, where inference can be solved efficiently using the junction algorithm [7]. However, learning

junction trees with tree-width greater than one is NP-complete [6] and tractable learning algorithms (e.g. [8]) often have constraints on both the tree-width and the maximum degree. Since graphs with large-degree nodes are important in modeling applications such as social networks, flight networks, and robotic localization, we are interested in finding a family of models that allow arbitrarily large degrees while being tractable for learning.

Beyond thin-junction trees, the family of sparse GGMs is also widely studied [9, 10]. These models are often estimated using methods such as graphical lasso (or $l$-1 regularization) [11, 12]. However, a sparse GGM (e.g. a grid) does not automatically lead to efficient algorithms for exact inference. Hence, we are interested in finding a family of models that are not only sparse but also have guaranteed efficient inference algorithms.

In this paper, we study the family of GGMs with small feedback vertex sets (FVSs), where an FVS is a set of nodes whose removal breaks all cycles [13]. The authors of [14] have demonstrated that the computation of exact means and variances for such a GGM can be accomplished, using message-passing algorithms with complexity $\mathcal{O}(k^2 n)$, where $k$ is the size of the FVS and $n$ is the total number of nodes. They have also presented results showing that for models with larger FVSs, approximate inference (obtained by replacing a full FVS by a pseudo-FVS) can work very well, with empirical evidence indicating that a pseudo-FVS of size $\mathcal{O}(\log n)$ gives excellent results. In Appendix A we will provide some additional analysis of inference for such models (including the computation of the partition function), but the main focus is maximum likelihood (ML) *learning* of models with FVSs of modest size, including identifying the nodes to include in the FVS.

In particular, we investigate two cases. In the first, all of the variables, including any to be included in the FVS are observed. We provide an algorithm for exact ML estimation that, regardless of the maximum degree, has complexity $\mathcal{O}(kn^2 + n^2 \log n)$ if the FVS nodes are identified in advance and polynomial complexity if the FVS is to be learned and of bounded size. Moreover, we provide an approximate and much faster greedy algorithm when the FVS is unknown *and* large. In the second case, the FVS nodes are taken to be latent variables. In this case, the structure learning problem corresponds to the (exact or approximate) decomposition of an inverse covariance matrix into the sum of a tree-structured matrix and a low-rank matrix. We propose an algorithm that iterates between two projections, which can also be interpreted as alternating *low-rank* corrections. We prove that even though the second projection is onto a highly non-convex set, it is carried out exactly, thanks to the properties of GGMs of this family. By carefully incorporating efficient inference into the learning steps, we can further reduce the complexity to $\mathcal{O}(kn^2 + n^2 \log n)$ per iteration. We also perform experiments using both synthetic data and real data of flight delays to demonstrate the modeling capacity with FVSs of various sizes. We show that empirically the family of GGMs of size $\mathcal{O}(\log n)$ strikes a good balance between the modeling capacity and efficiency.

**Related Work**   In the context of classification, the authors of [15] have proposed the tree augmented naive Bayesian model, where the class label variable itself can be viewed as a size-one observed FVS; however, this model does not naturally extend to include a larger FVS. In [16], a convex optimization framework is proposed to learn GGMs with latent variables, where conditioned on a small number of latent variables, the remaining nodes induce a sparse graph. In our setting with latent FVSs, we further require the sparse subgraph to have tree structure.

## 2   Preliminaries

Each undirected graphical model has an underlying graph $\mathcal{G} = (\mathcal{V}, \mathcal{E})$, where $\mathcal{V}$ denotes the set of vertices (nodes) and $\mathcal{E}$ the set of edges. Each node $s \in \mathcal{V}$ corresponds to a random variable $x_s$. When the random vector $\mathbf{x}_\mathcal{V}$ is jointly Gaussian, the model is a GGM with density function given by $p(\mathbf{x}) = \frac{1}{Z} \exp\{-\frac{1}{2} \mathbf{x}^T J \mathbf{x} + \mathbf{h}^T \mathbf{x}\}$, where $J$ is the *information matrix* or *precision matrix*, $\mathbf{h}$ is the *potential vector*, and $Z$ is the *partition function*. The parameters $J$ and $\mathbf{h}$ are related to the mean $\boldsymbol{\mu}$ and covariance matrix $\Sigma$ by $\boldsymbol{\mu} = J^{-1}\mathbf{h}$ and $\Sigma = J^{-1}$. The structure of the underlying graph is revealed by the sparsity pattern of $J$: there is an edge between $i$ and $j$ if and only if $J_{ij} \neq 0$.

Given samples $\{\mathbf{x}^i\}_{i=1}^s$ independently generated from an unknown distribution $q$ in the family $\mathcal{Q}$, the ML estimate is defined as $q_{\mathrm{ML}} = \arg\min_{q \in \mathcal{Q}} \sum_{i=1}^s \log q(\mathbf{x}^i)$. For Gaussian distributions, the empirical distribution is $\hat{p}(\mathbf{x}) = \mathcal{N}(\mathbf{x}; \hat{\boldsymbol{\mu}}, \hat{\Sigma})$, where the empirical mean $\hat{\boldsymbol{\mu}} = \frac{1}{s} \sum_{i=1}^s \mathbf{x}^i$ and the empirical covariance matrix $\hat{\Sigma} = \frac{1}{s} \sum_{i=1}^s \mathbf{x}^i (\mathbf{x}^i)^T - \hat{\boldsymbol{\mu}}\hat{\boldsymbol{\mu}}^T$. The Kullback-Leibler (K-L) divergence between two distributions $p$ and $q$ is defined as $D_{\mathrm{KL}}(p||q) = \int p(\mathbf{x}) \log \frac{p(\mathbf{x})}{q(\mathbf{x})} \mathrm{d}\mathbf{x}$. Without loss of generality, we assume in this paper the means are zero.

Tree-structured models are models whose underlying graphs do not have cycles. The ML estimate of a tree-structured model can be computed exactly using the Chow-Liu algorithm [2]. We use $\Sigma_{\mathrm{CL}} = \mathrm{CL}(\hat{\Sigma})$ and $\mathcal{E}_{\mathrm{CL}} = \mathrm{CL}_{\mathcal{E}}(\hat{\Sigma})$ to denote respectively the covariance matrix and the set of edges learned using the Chow-Liu algorithm where the samples have empirical covariance matrix $\hat{\Sigma}$.

## 3  Gaussian Graphical Models with Known FVSs

In this section we briefly discuss some of the ideas related to GGMs with FVSs of size $k$, where we will also refer to the nodes in the FVS as *feedback nodes*. An example of a graph and its FVS is given in Figure 1, where the full graph (Figure 1a) becomes a cycle-free graph (Figure 1b) if nodes 1 and 2 are removed, and thus the set $\{1, 2\}$ is an FVS.

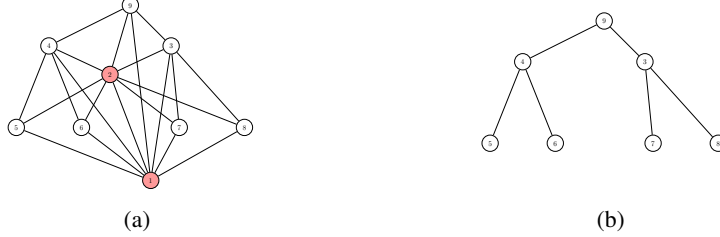

(a)                                                    (b)

Figure 1: A graph with an FVS of size 2. (a) Full graph; (b) Tree-structured subgraph after removing nodes 1 and 2

Graphs with small FVSs have been studied in various contexts. The authors of [17] have characterized the family of graphs with small FVSs and their obstruction sets (sets of forbidden minors). FVSs are also related to the "stable sets" in the study of tournaments [18].

Given a GGM with an FVS of size $k$ (where the FVS may or may not be given), the marginal means and variances $\boldsymbol{\mu}_i = \left(J^{-1}\mathbf{h}\right)_i$ and $\Sigma_{ii} = \left(J^{-1}\right)_{ii}$, for $\forall i \in \mathcal{V}$ can be computed *exactly* with complexity $\mathcal{O}(k^2 n)$ using the feedback message passing (FMP) algorithm proposed in [14], where standard BP is employed two times on the cycle-free subgraph among the non-feedback nodes while a special message-passing protocol is used for the FVS nodes. We provide a new algorithm in Appendix D, to compute $\det J$, the determinant of $J$, and hence the partition function of such a model with complexity $\mathcal{O}(k^2 n)$. The algorithm is described and proved in Appendix A.

An important point to note is that the complexity of these algorithms depends simply on the size $k$ and the number of nodes $n$. There is no loss in generality in assuming that the size-$k$ FVS $F$ is fully connected and each of the feedback nodes has edges to every non-feedback node. In particular, after re-ordering the nodes so that the elements of $F$ are the first $k$ nodes ($T = V \backslash F$ is the set of non-feedback nodes of size $n - k$), we have that $J = \begin{bmatrix} J_F & J_M^T \\ J_M & J_T \end{bmatrix} \succ 0$, where $J_T \succ 0$ corresponds to a tree-structured subgraph among the non-feedback nodes, $J_F \succ 0$ corresponds to a complete graph among the feedback nodes, and all entries of $J_M$ may be non-zero as long as $J_T - J_M J_F^{-1} J_M^T \succ 0$ (while $\Sigma = \begin{bmatrix} \Sigma_F & \Sigma_M^T \\ \Sigma_M & J_T \end{bmatrix} = J^{-1} \succ 0$ ). We will refer to the family of such models with a given FVS $F$ as $\mathcal{Q}_F$, and the class of models with some FVS of size at most $k$ as $\mathcal{Q}_k$.[1] If we are not explicitly given an FVS, though the problem of finding an FVS of minimal size is NP-complete, the authors of [19] have proposed an efficient algorithm with complexity $\mathcal{O}(\min\{m \log n, \ n^2\})$, where $m$ is the number of edges, that yields an FVS at most twice the minimum size (thus the inference complexity is increased only by a constant factor). However, the main focus of this paper, explored in the next section, is on *learning* models with small FVSs (so that when learned, the FVS is *known*). As we will see, the complexity of such algorithms is manageable. Moreover, as our experiments will demonstrate, for many problems, quite modestly sized FVSs suffice.

## 4  Learning GGMs with Observed or Latent FVS of Size $k$

In this section, we study the problem of recovering a GGM from $i.i.d.$ samples, where the feedback nodes are either observed or latent variables. If all nodes are observed, the empirical distribution

$\hat{p}(\mathbf{x}_F, \mathbf{x}_T)$ is parametrized by the empirical covariance matrix $\hat{\Sigma} = \begin{bmatrix} \hat{\Sigma}_F & \hat{\Sigma}_M^T \\ \hat{\Sigma}_M & \hat{\Sigma}_T \end{bmatrix}$. If the feedback

nodes are latent variables, the empirical distribution $\hat{p}(\mathbf{x}_T)$ has empirical covariance matrix $\hat{\Sigma}_T$. With a slight abuse of notation, for a set $A \subset \mathcal{V}$, we use $q(\mathbf{x}_A)$ to denote the marginal distribution of $\mathbf{x}_A$ under a distribution $q(\mathbf{x}_{\mathcal{V}})$.

## 4.1 When All Nodes Are Observed

When all nodes are observed, we have two cases: 1) When an FVS of size $k$ is given, we propose the *conditioned Chow-Liu algorithm,* which computes the *exact* ML estimate efficiently; 2) When no FVS is given *a priori*, we propose both an exact algorithm and a greedy approximate algorithm for computing the ML estimate.

### 4.1.1 Case 1: An FVS of Size $k$ Is Given.

When a size-$k$ FVS $F$ is given, the learning problem becomes solving

$$p_{\text{ML}}(\mathbf{x}_F, \mathbf{x}_T) = \underset{q(\mathbf{x}_F, \mathbf{x}_T) \in \mathcal{Q}_F}{\arg\min} D_{\text{KL}}(\hat{p}(\mathbf{x}_F, \mathbf{x}_T) \| q(\mathbf{x}_F, \mathbf{x}_T)). \tag{1}$$

This optimization problem is defined on a highly non-convex set $\mathcal{Q}_F$ with combinatorial structures: indeed, there are $(n - k)^{n-k-2}$ possible spanning trees among the subgraph induced by the non-feedback nodes. However, we are able to solve Problem (1) *exactly* using the conditioned Chow-Liu algorithm described in Algorithm 1.[2] The intuition behind this algorithm is that even though the entire graph is not tree, the subgraph induced by the non-feedback nodes (which corresponds to the distribution of the non-feedback nodes conditioned on the feedback nodes) has tree structure, and thus we can find the best tree among the non-feedback nodes using the Chow-Liu algorithm applied on the conditional distribution. To obtain a concise expression, we also exploit a property of Gaussian distributions: the conditional information matrix (the information matrix of the conditional distribution) is simply a submatrix of the whole information matrix. In Step 1 of Algorithm 1, we compute the conditional covariance matrix using the Schur complement, and then in Step 2 we use the Chow-Liu algorithm to obtain the best approximate $\Sigma_{\text{CL}}$ (whose inverse is tree-structured). In Step 3, we match exactly the covariance matrix among the feedback nodes and the covariance matrix between the feedback nodes and the non-feedback nodes. For the covariance matrix among the non-feedback nodes, we add the matrix subtracted in Step 1 back to $\Sigma_{\text{CL}}$. Proposition 1 states the correctness and the complexity of Algorithm 1. Its proof included in Appendix B. We denote the output covariance matrix of this algorithm as $\text{CCL}(\hat{\Sigma})$.

---

**Algorithm 1** The conditioned Chow-Liu algorithm

---

**Input:** $\hat{\Sigma} \succ 0$ and an FVS $F$
**Output:** $\mathcal{E}_{\text{ML}}$ and $\Sigma_{\text{ML}}$

1. Compute the conditional covariance matrix $\hat{\Sigma}_{T|F} = \hat{\Sigma}_T - \hat{\Sigma}_M \hat{\Sigma}_F^{-1} \hat{\Sigma}_M^T$ .

2. Let $\Sigma_{\text{CL}} = \text{CL}(\hat{\Sigma}_{T|F})$ and $\mathcal{E}_{\text{CL}} = \text{CL}_{\mathcal{E}}(\hat{\Sigma}_{T|F})$.

3. $\mathcal{E}_{\text{ML}} = \mathcal{E}_{\text{CL}}$ and $\Sigma_{\text{ML}} = \begin{bmatrix} \hat{\Sigma}_F & \hat{\Sigma}_M^T \\ \hat{\Sigma}_M & \Sigma_{\text{CL}} + \hat{\Sigma}_M \hat{\Sigma}_F^{-1} \hat{\Sigma}_M^T \end{bmatrix}$.

---

**Proposition 1.** *Algorithm 1 computes the ML estimate $\Sigma_{ML}$ and $\mathcal{E}_{ML}$, exactly with complexity $\mathcal{O}(kn^2 + n^2 \log n)$. In addition, all the non-zero entries of $J_{ML} \triangleq \Sigma_{ML}^{-1}$ can be computed with extra complexity $\mathcal{O}(k^2 n)$.*

### 4.1.2 Case 2: The FVS Is to Be Learned

Structure learning becomes more computationally involved when the FVS is unknown. In this subsection, we present both exact and approximate algorithms for learning models with FVS of size no larger than $k$ (i.e., in $\mathcal{Q}_k$). For a fixed empirical distribution $\hat{p}(\mathbf{x}_F, \mathbf{x}_T)$, we define $d(F)$, a set function of the FVS $F$ as the minimum value of (1), i.e.,

$$d(F) = \min_{q(\mathbf{x}_F, \mathbf{x}_T) \in \mathcal{Q}_F} D_{\text{KL}}(\hat{p}(\mathbf{x}_F, \mathbf{x}_T) || q(\mathbf{x}_F, \mathbf{x}_T)). \tag{2}$$

When the FVS is unknown, the ML estimate can be computed exactly by enumerating all possible $\binom{n}{k}$ FVSs of size $k$ to find the $F$ that minimizes $d(F)$. Hence, the exact solution can be obtained with complexity $\mathcal{O}(n^{k+2}k)$, which is polynomial in $n$ for fixed $k$. However, as our empirical results suggest, choosing $k = \mathcal{O}(\log(n))$ works well, leading to quasi-polynomial complexity even for this exact algorithm. That observation notwithstanding, the following greedy algorithm (Algorithm 2), which, at each iteration, selects the single best node to add to the current set of feedback nodes, has polynomial complexity for arbitrarily large FVSs. As we will demonstrate, this greedy algorithm works extremely well in practice.

---
**Algorithm 2** Selecting an FVS by a greedy approach
---
    **Initialization:** $F_0 = \emptyset$
    **For** $t = 1$ **to** $k$,
$$k_t^* = \operatorname*{arg\,min}_{k \in V \setminus F_{t-1}} d(F_{t-1} \cup \{k\}), \ F_t = F_{t-1} \cup \{k_t^*\}$$

---

### 4.2 When the FVS Nodes Are Latent Variables

When the feedback nodes are latent variables, the marginal distribution of observed variables (the non-feedback nodes in the true model) has information matrix $\tilde{J}_T = \hat{\Sigma}_T^{-1} = J_T - J_M J_F^{-1} J_M^T$. If the exact $\tilde{J}_T$ is known, the learning problem is equivalent to decomposing a given inverse covariance matrix $\tilde{J}_T$ into the sum of a tree-structured matrix $J_T$ and a rank-$k$ matrix $-J_M J_F^{-1} J_M^T$.[3] In general, use the ML criterion

$$q_{\text{ML}}(\mathbf{x}_F, \mathbf{x}_T) = \arg \min_{q(\mathbf{x}_F, \mathbf{x}_T) \in Q_F} D_{\text{KL}}(\hat{p}(\mathbf{x}_T) || q(\mathbf{x}_T)), \tag{3}$$

where the optimization is over all nodes (latent and observed) while the K-L divergence in the objective function is defined on the marginal distribution of the observed nodes only.

We propose *the latent Chow-Liu algorithm,* an alternating projection algorithm that is a variation of the EM algorithm and can be viewed as an instance of the majorization-minimization algorithm. The general form of the algorithm is as follows:

1. Project onto the empirical distribution:
$$\hat{p}^{(t)}(\mathbf{x}_F, \mathbf{x}_T) = \hat{p}(\mathbf{x}_T)q^{(t)}(\mathbf{x}_F | \mathbf{x}_T).$$

2. Project onto the best fitting structure on all variables:
$$q^{(t+1)}(\mathbf{x}_F, \mathbf{x}_T) = \arg \min_{q(\mathbf{x}_F, \mathbf{x}_T) \in \mathcal{Q}_F} D_{\text{KL}}(\hat{p}^{(t)}(\mathbf{x}_F, \mathbf{x}_T) || q(\mathbf{x}_F, \mathbf{x}_T)).$$

In the first projection, we obtain a distribution (on both observed and latent variables) whose marginal (on the observed variables) matches exactly the empirical distribution while maintaining the conditional distribution (of the latent variables given the observed ones). In the second projection we compute a distribution (on all variables) in the family considered that is the closest to the distribution obtained in the first projection. We found that among various EM type algorithms, this formulation is the most revealing for our problems because it clearly relates the second projection to the scenario where an FVS $F$ is both observed and known (Section 4.1.1). Therefore, we are able to compute the second projection *exactly* even though the graph structure is *unknown* (which allows *any* tree structure among the observed nodes). Note that when the feedback nodes are latent, we do

not need to select the FVS since it is simply the set of latent nodes. This is the source of the simplification when we use latent nodes for the FVS: We have no search of sets of observed variables to include in the FVS.

---

**Algorithm 3** The latent Chow-Liu algorithm

---

**Input:** the empirical covariance matrix $\hat{\Sigma}_T$

**Output:** information matrix $J = \begin{bmatrix} J_F & J_M^T \\ J_M & J_T \end{bmatrix}$, where $J_T$ is tree-structured

1. Initialization: $J^{(0)} = \begin{bmatrix} J_F^{(0)} & \left(J_M^{(0)}\right)^T \\ J_M^{(0)} & J_T^{(0)} \end{bmatrix}$.

2. Repeat for $t = 1, 2, 3, \ldots$:

   (a) **P1**: Project to the empirical distribution:
   $$\hat{J}^{(t)} = \begin{bmatrix} J_F^{(t)} & (J_M^{(t)})^T \\ J_M^{(t)} & \left(\hat{\Sigma}_T\right)^{-1} + J_M^{(t)}(J_F^{(t)})^{-1}(J_M^{(t)})^T \end{bmatrix}. \text{ Define } \hat{\Sigma}^{(t)} = \left(\hat{J}^{(t)}\right)^{-1}.$$

   (b) **P2:** Project to the best fitting structure:
   $$\Sigma^{(t+1)} = \begin{bmatrix} \hat{\Sigma}_F^{(t)} & \left(\hat{\Sigma}_M^{(t)}\right)^T \\ \hat{\Sigma}_M^{(t)} & \mathrm{CL}(\hat{\Sigma}_{T|F}^{(t)}) + \hat{\Sigma}_M^{(t)}\left(\hat{\Sigma}_F^{(t)}\right)^{-1}\left(\hat{\Sigma}_M^{(t)}\right)^T \end{bmatrix} = \mathrm{CCL}(\hat{\Sigma}^{(t)}),$$
   where $\hat{\Sigma}_{T|F}^{(t)} = \hat{\Sigma}_T^{(t)} - \hat{\Sigma}_M^{(t)}\left(\hat{\Sigma}_F^{(t)}\right)^{-1}\left(\hat{\Sigma}_M^{(t)}\right)^T$. Define $J^{(t+1)} = \left(\Sigma^{(t+1)}\right)^{-1}$.

---

In Algorithm 3 we summarize the latent Chow-Liu algorithm specialized for our family of GGMs, where both projections have exact closed-form solutions and exhibit complementary structure—one using the covariance and the other using the information parametrization. In projection **P1**, three blocks of the information matrix remain the same; In projection **P2**, three blocks of the covariance matrix remain the same.

The two projections in Algorithm 3 can also be interpreted as alternating *low-rank* corrections :
indeed,

$$\text{In P1} \qquad \hat{J}^{(t)} \quad = \begin{bmatrix} \mathbf{0} & \mathbf{0} \\ \mathbf{0} & \left(\hat{\Sigma}_T\right)^{-1} \end{bmatrix} + \begin{bmatrix} J_F^{(t)} \\ J_M^{(t)} \end{bmatrix} \left(J_F^{(t)}\right)^{-1} \begin{bmatrix} J_F^{(t)} & \left(J_M^{(t)}\right)^T \end{bmatrix},$$

$$\text{and in P2} \quad \Sigma^{(t+1)} \quad = \begin{bmatrix} \mathbf{0} & \mathbf{0} \\ \mathbf{0} & \mathrm{CL}(\hat{\Sigma}_{T|F}) \end{bmatrix} + \begin{bmatrix} \hat{\Sigma}_F^{(t)} \\ \hat{\Sigma}_M^{(t)} \end{bmatrix} \left(\hat{\Sigma}_F^{(t)}\right)^{-1} \begin{bmatrix} \hat{\Sigma}_F^{(t)} & \left(\hat{\Sigma}_M^{(t)}\right)^T \end{bmatrix},$$

where the second terms of both expressions are of low-rank when the size of the latent FVS is small. This formulation is the most intuitive and simple, but a naive implementation of Algorithm 3 has complexity $\mathcal{O}(n^3)$ per iteration, where the bottleneck is inverting full matrices $\hat{J}^{(t)}$ and $\Sigma^{(t+1)}$. By carefully incorporating the inference algorithms into the projection steps, we are able to further exploit the power of the models and reduce the per-iteration complexity to $\mathcal{O}(kn^2 + n^2 \log n)$, which is the same as the complexity of the conditioned Chow-Liu algorithm alone. We have the following proposition.

**Proposition 2.** *Using Algorithm 3, the objective function of* (3) *decreases with the number of iterations, i.e.,* $D_{KL}(\mathcal{N}(0, \hat{\Sigma}_T) || \mathcal{N}(0, \Sigma_T^{(t+1)})) \le \mathcal{N}(0, \hat{\Sigma}_T) || \mathcal{N}(0, \Sigma_T^{(t)}))$. *Using an accelerated version of Algorithm 3, the complexity per iteration is* $\mathcal{O}(kn^2 + n^2 \log n)$.

Due to the page limit, we defer the description of the accelerated version (*the accelerated latent Chow-Liu algorithm*) and the proof of Proposition 2 to Appendix C. In fact, we never need to explicitly invert the empirical covariance matrix $\hat{\Sigma}_T$ in the accelerated version.

As a rule of thumb, we often use the spanning tree obtained by the standard Chow-Liu algorithm as an initial tree among the observed nodes. But note that **P2** involves solving a combinatorial problem exactly, so the algorithm is able to jump among different graph structures which reduces the chance

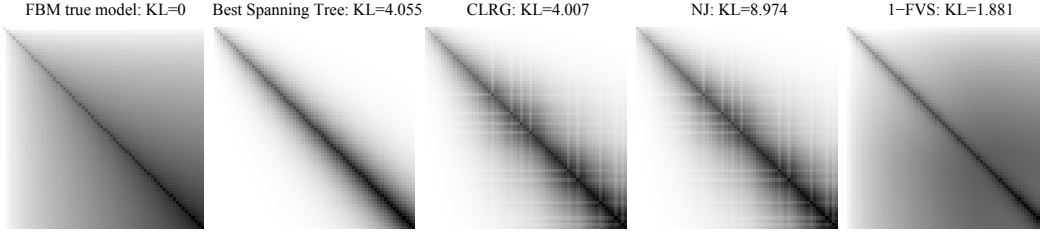

Figure 2: From left to right: 1) The true model (fBM with 64 time samples); 2) The best spanning tree; 3) The latent tree learned using the CLRG algorithm in [21]; 4) The latent tree learned using the NJ algorithm in [21]; 5) The model with a size-one latent FVS learned using Algorithm 3. The gray scale is normalized for visual clarity.

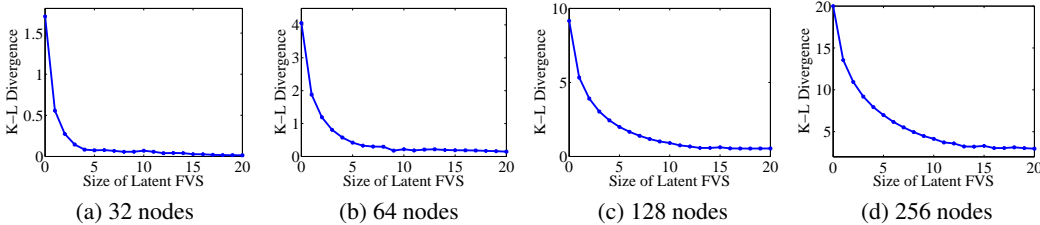

Figure 3: The relationship between the K-L divergence and the latent FVS size. All models are learned using Algorithm 3 with 40 iterations.

of getting stuck at a bad local minimum and gives us much more flexibility in initializing graph structures. In the experiments, we will demonstrate that Algorithm 3 is not sensitive to the initial graph structure.

## 5   Experiments

In this section, we present experimental results for learning GGMs with small FVSs, observed or latent, using both synthetic data and real data of flight delays.

**Fractional Brownian Motion: Latent FVS**   We consider a fractional Brownian motion (fBM) with Hurst parameter $H = 0.2$ defined on the time interval $(0, 1]$. The covariance function is $\Sigma(t_1, t_2) = \frac{1}{2}(|t_1|^{2H} + |t_2|^{2H} - |t_1 - t_2|^{2H})$. Figure 2 shows the covariance matrices of approximate models using spanning trees (learned by the Chow-Liu algorithm), latent trees (learned by the CLRG and NJ algorithms in [21]) and our latent FVS model (learned by Algorithm 3) using 64 time samples (nodes). We can see that in the spanning tree the correlation decays quickly (in fact exponentially) with distance, which models the fBM poorly. The latent trees that are learned exhibit blocky artifacts and have little or no improvement over the spanning tree measured in the K-L divergence. In Figure 3, we plot the K-L divergence (between the true model and the learned models using Algorithm 3) versus the size of the latent FVSs for models with 32, 64, 128, and 256 time samples respectively. For these models, we need about 1, 3, 5, and 7 feedback nodes respectively to reduce the K-L divergence to 25% of that achieved by the best spanning tree model. Hence, we speculate that empirically $k = \mathcal{O}(\log n)$ is a proper choice of the size of the latent FVS. We also study the sensitivity of Algorithm 3 to the initial graph structure. In our experiments, for different initial structures, Algorithm 3 converges to the same graph structures (that give the K-L divergence as shown in Figure 3) within three iterations.

**Performance of the Greedy Algorithm: Observed FVS**   In this experiment, we examine the performance of the greedy algorithm (Algorithm 2) when the FVS nodes are observed. For each run, we construct a GGM that has 20 nodes and an FVS of size three as the true model. We first generate a random spanning tree among the non-feedback nodes. Then the corresponding information matrix $J$ is also randomly generated: non-zero entries of $J$ are drawn *i.i.d.* from the uniform distribution $U[-1, 1]$ with a multiple of the identity matrix added to ensure $J \succ 0$. From each generated GGM, we draw 1000 samples and use Algorithm 2 to learn the model. For 100 runs that we have performed, we recover the true graph structures successfully. Figure 4 shows the graphs (and the K-L divergence) obtained using the greedy algorithm for a typical run. We can see that we have the most divergence reduction (from 12.7651 to 1.3832) when the first feedback node is selected. When the size of the FVS increases to three (Figure 4e), the graph structure is recovered correctly.

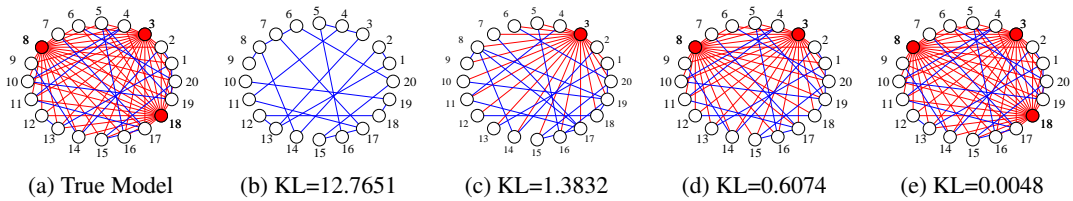

| (a) True Model | (b) KL=12.7651 | (c) KL=1.3832 | (d) KL=0.6074 | (e) KL=0.0048 |

Figure 4: Learning a GGM using Algorithm 2. The thicker blue lines represent the edges among the non-feedback nodes and the thinner red lines represent other edges. (a) True model; (b) Tree-structured model (0-FVS) learned from samples; (c) 1-FVS model; (d) 2-FVS model; (e) 3-FVS model.

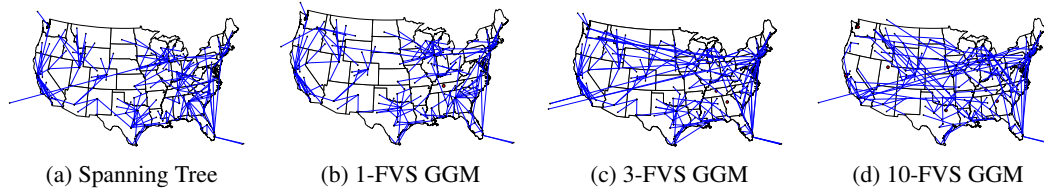

| (a) Spanning Tree | (b) 1-FVS GGM | (c) 3-FVS GGM | (d) 10-FVS GGM |

Figure 5: GGMs for modeling flight delays. The red dots denote selected feedback nodes and the blue lines represent edges among the non-feedback nodes (other edges involving the feedback nodes are omitted for clarity).

**Flight Delay Model: Observed FVS**   In this experiment, we model the relationships among airports for flight delays. The raw dataset comes from RITA of the Bureau of Transportation Statistics. It contains flight information in the U.S. from 1987 to 2008 including information such as scheduled departure time, scheduled arrival time, departure delay, arrival delay, cancellation, and reasons for cancellation for all domestic flights in the U.S. We want to model how the flight delays at different airports are related to each other using GGMs. First, we compute the average departure delay for each day and each airport (of the top 200 busiest airports) using data from the year 2008. Note that the average departure delays does not directly indicate whether an airport is one of the major airports that has heavy traffic. It is interesting to see whether major airports (especially those notorious for delays) correspond to feedback nodes in the learned models. Figure 5a shows the best tree-structured graph obtained by the Chow-Liu algorithms (with input being the covariance matrix of the average delay). Figure 5b–5d show the GGMs learned using Algorithm 2. It is interesting that the first node selected is Nashville (BNA), which is not one of the top "hubs" of the air system. The reason is that much of the statistical relationships related to those hubs are approximated well enough, when we consider a 1-FVS approximation, by a spanning tree (excluding BNA) and it is the breaking of the cycles involving BNA that provide the most reduction in K-L divergence over a spanning tree. Starting with the next node selected in our greedy algorithm, we begin to see hubs being chosen. In particular, the first ten airports selected in order are: BNA, Chicago, Atlanta, Oakland, Newark, Dallas, San Francisco, Seattle, Washington DC, Salt Lake City. Several major airports on the coasts (e.g., Los Angeles and JFK) are not selected, as their influence on delays at other domestic airports is well-captured with a tree structure.

## 6   Future Directions

Our experimental results demonstrate the potential of these algorithms, and, as in the work [14], suggests that choosing FVSs of size $\mathcal{O}(\log n)$ works well, leading to algorithms which can be scaled to large problems. Providing theoretical guarantees for this scaling (e.g., by specifying classes of models for which such a size FVS provides asymptotically accurate models) is thus a compelling open problem. In addition, incorporating complexity into the FVS-order problem (e.g., as in AIC or BIC) is another direction we are pursuing. Moreover, we are also working towards extending our results to the non-Gaussian settings.

### Acknowledgments

This research was supported in part by AFOSR under Grant FA9550-12-1-0287.

## Footnotes

[1]In general a graph does not have a unique FVS. The family of graphs with FVSs of size $k$ includes all graphs where there *exists* an FVS of size $k$.

[2] Note that the conditioned Chow-Liu algorithm here is different from other variations of the Chow-Liu algorithm such as in [20] where the extensions are to enforce the inclusion or exclusion of a set of edges.

[3]It is easy to see that different models having the same $J_M J_F^{-1} J_M$ cannot be distinguished using the samples, and thus without loss of generality we can assume $J_F$ is normalized to be the identify matrix in the final solution.

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
