[Supplementary Material]

# Appendix of "Learning Gaussian Graphical Models with Observed or Latent FVSs"

## A    Computing the Partition Function of GGMs in $\mathcal{Q}_F$

In Section 3 of the paper, we have stated that given the information matrix $J$ of a GGM with an FVS of size $k$, we can compute $\det J$ and hence the partition function using a message-passing algorithm with complexity $\mathcal{O}(k^2 n)$. This algorithm is inspired by the FMP algorithm developed in [14] and is described in Algorithm 4.

---

**Algorithm 4** Computing the partition function when an FVS is given

---

**Input:** an FVS $F$ of size $k$ and an $n \times n$ information matrix $J = \begin{bmatrix} J_F & J_M^T \\ J_M & J_T \end{bmatrix}$, where $J_T$ has tree structure $\mathcal{T}$ with edge set $\mathcal{E}_T$.

**Output:** $\det J$

1. Run standard Gaussian BP on $\mathcal{T}$ with information matrix $J_T$ to obtain $P_{ii}^{\mathcal{T}} = \left(J_T^{-1}\right)_{ii}$ for all $i \in T$, $P_{ij}^{\mathcal{T}} = (J_T^{-1})_{ij}$ for all $(i,j) \in \mathcal{E}_T$, and $(\mathbf{g}^p)_i = (J_T^{-1}\mathbf{h}^p)_i$ for all $i \in T$ and $p \in F$, where $\mathbf{h}^p$ is the column of $J_M$ corresponding to node $p$.

2. Compute $\hat{J}_F$ with

$$\left(\hat{J}_F\right)_{pq} = J_{pq} - \sum_{j \in \mathcal{N}(p) \cap T} J_{pj} g_j^q, \ \forall \, p, q \in F$$

3. Compute $\det \hat{J}_F$, the determinant of $\hat{J}_F$.

4. Output

$$\det J = \left( \prod_{(i,j) \in \mathcal{E}_T} \frac{P_{ii}^{\mathcal{T}} P_{jj}^{\mathcal{T}} - \left(P_{ij}^{\mathcal{T}}\right)^2}{P_{ii}^{\mathcal{T}} P_{jj}^{\mathcal{T}}} \prod_{i \in \mathcal{V}} P_{ii}^{\mathcal{T}} \right)^{-1} \det \hat{J}_{\mathcal{F}}.$$

---

We state the correctness and the computational complexity of Algorithm 4 in Proposition 3.

**Proposition 3.** *Algorithm 4 computes $\det J$ exactly and the computational complexity is $\mathcal{O}(k^2 n)$.*

Before giving the proof for Proposition 3, we first prove Lemma 1.

**Lemma 1.** *If the information matrix $J \succ 0$ has tree structure $\mathcal{T} = (\mathcal{V}, \mathcal{E})$, then we have*

$$\det (J)^{-1} = \prod_{i \in \mathcal{V}} P_{ii} \prod_{(i,j) \in \mathcal{E}} \frac{P_{ii} P_{jj} - P_{ij}^2}{P_{ii} P_{jj}}, \tag{4}$$

*where $P = J^{-1}$.*

*Proof.* WLOG, we assume the means are zero. For any tree-structured distribution $p(\mathbf{x})$ with underlying tree $\mathcal{T}$, we have the following factorization:

$$p(\mathbf{x}) = \prod_{i \in \mathcal{V}} p(x_i) \prod_{(i,j) \in \mathcal{E}_T} \frac{p(x_i, x_j)}{p(x_i) p(x_j)}. \tag{5}$$

For a GGM of $n$ nodes, the joint distribution, the singleton marginal distributions, and the pairwise marginal distributions can be expressed as follows.

$$p(\mathbf{x}) = \frac{1}{(2\pi)^{\frac{n}{2}} (\det J)^{-\frac{1}{2}}} \exp\{-\frac{1}{2}\mathbf{x}^T J \mathbf{x}\}$$

$$p(x_i) = \frac{1}{(2\pi)^{\frac{1}{2}} P_{ii}^{\frac{1}{2}}} \exp\{-\frac{1}{2}\mathbf{x}^T P_{ii}^{-1} \mathbf{x}\}$$

$$p(x_i, x_j) = \frac{1}{2\pi \left(\det \begin{bmatrix} P_{ii} & P_{ij} \\ P_{ji} & P_{jj} \end{bmatrix}\right)^{\frac{1}{2}}} \exp\{-\frac{1}{2}\mathbf{x}^T \begin{bmatrix} P_{ii} & P_{ij} \\ P_{ji} & P_{jj} \end{bmatrix}^{-1} \mathbf{x}\}.$$

Matching the normalization factors using (5), we obtain

$$\det (J)^{-1} = \prod_{i\in\mathcal{V}} P_{ii} \prod_{(i,j)\in\mathcal{E}} \frac{\det \begin{bmatrix} P_{ii} & P_{ij} \\ P_{ji} & P_{jj} \end{bmatrix}}{P_{ii}P_{jj}}. \tag{6}$$

$$= \prod_{i\in\mathcal{V}} P_{ii} \prod_{(i,j)\in\mathcal{E}} \frac{P_{ii}P_{jj} - P_{ij}^2}{P_{ii}P_{jj}} \tag{7}$$

$\square$

Now we proceed to prove Proposition 3.

*Proof.* First, we show that $\hat{J}_F$ computed in Step 2 of Algorithm 4 equals $J_F - J_M^T J_T^{-1} J_M$. We have

$$\begin{bmatrix} \mathbf{g}^1 & \mathbf{g}^2 & \cdots & \mathbf{g}^k \end{bmatrix} = J_T^{-1} \begin{bmatrix} \mathbf{h}^1 & \mathbf{h}^2 & \cdots & \mathbf{h}^k \end{bmatrix} = J_T^{-1} J_M$$

from the definition in Step 1. From Step 3, we can get

$$\begin{aligned} \hat{J}_F &= J_F - \begin{bmatrix} \mathbf{g}^1 & \mathbf{g}^2 & \cdots & \mathbf{g}^k \end{bmatrix}^T J_T \begin{bmatrix} \mathbf{g}^1 & \mathbf{g}^2 & \cdots & \mathbf{g}^k \end{bmatrix} \\ &= J_F - \left(J_T^{-1} J_M\right)^T J_T \left(J_T^{-1} J_M\right) \\ &= J_F - J_M^T J_T^{-1} J_M. \end{aligned} \tag{8}$$

Hence,

$$\begin{aligned} \det J &= \det \left(\begin{bmatrix} I & -J_M^T J_T^{-1} \\ \mathbf{0} & I \end{bmatrix}\right) \det \left(\begin{bmatrix} J_F & J_M^T \\ J_M & J_T \end{bmatrix}\right) \det \left(\begin{bmatrix} I & \mathbf{0} \\ -J_T^{-1} J_M & I \end{bmatrix}\right) \\ &= \det \left(\begin{bmatrix} I & -J_M^T J_T^{-1} \\ \mathbf{0} & I \end{bmatrix} \begin{bmatrix} J_F & J_M^T \\ J_M & J_T \end{bmatrix} \begin{bmatrix} I & \mathbf{0} \\ -J_T^{-1} J_M & I \end{bmatrix}\right) \\ &= \det \begin{bmatrix} J_F - J_M^T J_T^{-1} J_M & \mathbf{0} \\ \mathbf{0} & J_T \end{bmatrix} \\ &= \left(\det \hat{J}_F\right) \times (\det J_T), \end{aligned} \tag{9}$$

From Lemma 1 to follow, we have

$$\det (J_T)^{-1} = \prod_{i\in\mathcal{V}} P_{ii}^{\mathcal{T}} \prod_{(i,j)\in\mathcal{E}_T} \frac{P_{ii}^{\mathcal{T}} P_{jj}^{\mathcal{T}} - \left(P_{ij}^{\mathcal{T}}\right)^2}{P_{ii}^{\mathcal{T}} P_{jj}^{\mathcal{T}}}. \tag{10}$$

Hence, we have proved the correctness of the algorithm. Now we calculate the complexity. The first step of Algorithm 4 has complexity $\mathcal{O}(n - k)$ using BP. Step 2 takes $\mathcal{O}\left(k^2(n - k)\right)$ and the complexity of Step 3 is $\mathcal{O}(k^3)$. Finally the complexity of Step 4 is $\mathcal{O}(n)$ since $\mathcal{T}$ is a tree. The total complexity is thus $\mathcal{O}(k^2 n)$. This completes the proof for Proposition 3.

$\square$

Note that if the FVS is not given, we can use the factor-2 approximate algorithm in [19] to obtain an FVS of size at most twice the minimum size with complexity $\mathcal{O}(\min\{m \log n, \ n^2\})$, where $m$ is the number of edges.

# B  Proof for Proposition 1

## B.1  Preliminaries

Proposition 1 states that Algorithm 1 computes the ML estimate with covariance $\Sigma_{\mathrm{ML}}$ (together with $\mathcal{E}_{\mathrm{ML}}$, the set of edges among the non-feedback nodes) exactly with complexity $\mathcal{O}(kn^2 + n^2 \log n)$, and that $J_{\mathrm{ML}} \triangleq \Sigma_{\mathrm{ML}}^{-1}$ can be computed with additional complexity $\mathcal{O}(k^2 n)$.

First, we define the following information quantities:

1. The entropy $H_{p_{\mathbf{x}}}(\mathbf{x}) = -\int_{\mathbf{x}} p_{\mathbf{x}}(\mathbf{x}) \log p_{\mathbf{x}}(\mathbf{x}) \mathrm{d}\mathbf{x}$

2. The conditional entropy $H_{p_{\mathbf{x},\mathbf{y}}}(\mathbf{x}|\mathbf{y}) = -\int_{\mathbf{x},\mathbf{y}} p_{\mathbf{x},\mathbf{y}}(\mathbf{x},\mathbf{y}) \log p_{\mathbf{x}|\mathbf{y}}(\mathbf{x}|\mathbf{y}) \mathrm{d}\mathbf{x}\mathrm{d}\mathbf{y}$

3. The mutual information $I_{p_{\mathbf{x},\mathbf{y}}}(\mathbf{x};\mathbf{y}) = \int_{\mathbf{x},\mathbf{y}} p_{\mathbf{x},\mathbf{y}}(\mathbf{x},\mathbf{y}) \log \frac{p(\mathbf{x})p(\mathbf{y})}{p(\mathbf{x},\mathbf{y})} \mathrm{d}\mathbf{x}\mathrm{d}\mathbf{y}$

4. The conditional mutual information
$$I_{p_{\mathbf{x},\mathbf{y},\mathbf{z}}}(\mathbf{x};\mathbf{y}|\mathbf{z}) = \int_{\mathbf{x},\mathbf{y},\mathbf{z}} p_{\mathbf{x},\mathbf{y},\mathbf{z}}(\mathbf{x},\mathbf{y},\mathbf{z}) \log \frac{p(\mathbf{x},\mathbf{y}|\mathbf{z})}{p(\mathbf{x}|\mathbf{z})p(\mathbf{y}|\mathbf{z})} \mathrm{d}\mathbf{x}\mathrm{d}\mathbf{y}$$

5. The conditional K-L divergence: $D(\hat{p}_{\mathbf{x}|\mathbf{y}}||q_{\mathbf{x}|\mathbf{y}}|\hat{p}_{\mathbf{y}}) \triangleq D(\hat{p}_{\mathbf{x},\mathbf{y}}||q_{\mathbf{x}|\mathbf{y}}\hat{p}_y)$.

The (conditional) K-L divergence is always nonnegative. It is zero if and only if the two distributions are the same (almost everywhere). When there is no confusion, the subscripts in the distributions are often omitted, e.g., $I_{p_{\mathbf{x},\mathbf{y}}}(\mathbf{x};\mathbf{y})$ written as $I_p(\mathbf{x};\mathbf{y})$. With a slight abuse of notation, l we use $p(\mathbf{x}_F)$ to denote the marginal distribution of $\mathbf{x}_F$ under the joint distribution $p(\mathbf{x})$, and similarly $p(\mathbf{x}_T|\mathbf{x}_F)$ to denote the conditional distribution of $\mathbf{x}_T$ given $\mathbf{x}_F$ under the joint distribution $p(\mathbf{x})$.

The standard Chow-Liu Algorithm for GGMs is summarized in 5. The complexity is $\mathcal{O}(n^2 \log n)$. Note that in Step 3, for a fixed $i$, for any $(i,j) \notin \mathcal{E}_T$, $\Sigma_{ij}$ can be computed following a topological order of with $i$ being the root. Hence, by book-keeping the computed products along the paths, the complexity of computing each $\Sigma_{ij}$ is $\mathcal{O}(1)$.

---

**Algorithm 5** the Chow-Liu Algorithm for GGMs

**Input:** the empirical covariance matrix $\hat{\Sigma}$
**Output:** $\Sigma_{\mathrm{CL}}$ and $\mathcal{E}_{\mathrm{CL}}$

1. Compute the correlation coefficients $\rho_{ij} = \frac{\hat{\Sigma}_{ij}}{\sqrt{\hat{\Sigma}_{ii}\hat{\Sigma}_{jj}}}$

2. Find an MST (maximum weight spanning tree) of the complete graph with weights $|\rho_{ij}|$ for edge $(i,j)$. The edge set of the tree is denoted as $\mathcal{E}_T$.

3. For all $i \in \mathcal{V}$, $(\Sigma_{\mathrm{CL}})_{ii} = \hat{\Sigma}_{ii}$; for $(i,j) \in \mathcal{E}_T$, $(\Sigma_{\mathrm{CL}})_{ij} = \hat{\Sigma}_{ij}$; for $(i,j) \notin \mathcal{E}_T$, $(\Sigma_{\mathrm{CL}})_{ij} = \sqrt{\Sigma_{ii}\Sigma_{jj}} \prod_{(l,k)\in\mathrm{Path}(i,j)} \rho_{lk}$, where $\mathrm{Path}(i,j)$ is the set of edges on the unique path between $i$ and $j$ in the spanning tree.

---

## B.2  Lemmas

Lemma 2 is a well-known result stated without proof.

**Lemma 2.** *The p.d.f. of a tree-structured model $\mathcal{T} = (\mathcal{V}, \mathcal{E})$ can be factorized according to either of the following two equations:*

1. *$p(\mathbf{x}) = p(\mathrm{x}_r) \prod_{i \in \mathcal{V} \backslash r} p(\mathrm{x}_i | \mathrm{x}_{\pi(i)})$, where $r$ is an arbitrary node selected as the root and $\pi(i)$ is the unique parent of node $i$ in the tree rooted at $r$.*

2. $p(\mathbf{x}) = \prod_{i \in \mathcal{V}} p(\mathbf{x}_i) \prod_{(i,j) \in \mathcal{E}} \frac{p(\mathbf{x}_i, \mathbf{x}_j)}{p(\mathbf{x}_i)p(\mathbf{x}_j)}$.

For a given $F$ and a fixed tree $\mathcal{T}$ with edge set $\mathcal{E}_T$ among the non-feedback nodes, Lemma 3 gives a closed form solution that minimizes the K-L divergence.

**Lemma 3.**

$$\min_{q \in \mathcal{Q}_{F,\mathcal{T}}} D_{KL}(\hat{p}||q) = -H_{\hat{p}}(\mathbf{x}) + H_{\hat{p}}(\mathbf{x}_F) + \sum_{i \in \mathcal{V} \setminus F} H_{\hat{p}}(\mathbf{x}_i|\mathbf{x}_F) - \sum_{(i,j) \in \mathcal{E}_{\mathcal{T}}} I_{\hat{p}}(\mathbf{x}_i; \mathbf{x}_j|\mathbf{x}_F), \quad (11)$$

*where $\mathcal{Q}_{F,\mathcal{T}}$ is the set of distributions defined on a graph with a given FVS $F$ and a given spanning tree $\mathcal{T}$ among the non-feedback nodes. The minimum K-L divergence is obtained if and only if: 1) $q(\mathbf{x}_F) = \hat{p}(\mathbf{x}_F)$; 2) $q(\mathbf{x}_F, x_i, x_j) = \hat{p}(\mathbf{x}_F, x_i, x_j)$ for any $(i,j) \in \mathcal{E}_{\mathcal{T}}$.*

*Proof.* With fixed $F$ and $\mathcal{T}$,

$$
\begin{aligned}
D_{\mathrm{KL}}(\hat{p}||q) &= \int \hat{p}(\mathbf{x}) \log \frac{\hat{p}(\mathbf{x})}{q(\mathbf{x})} \mathrm{d}\mathbf{x} \\
&= -H_{\hat{p}}(\mathbf{x}) - \int \hat{p}(\mathbf{x}) \log q(\mathbf{x}) \mathrm{d}\mathbf{x} \\
&= -H_{\hat{p}}(\mathbf{x}) - \int \hat{p}(\mathbf{x}) \log\left(q(\mathbf{x}_F)q(\mathbf{x}_T|\mathbf{x}_F)\right) \mathrm{d}\mathbf{x} \\
&\overset{(a)}{=} -H_{\hat{p}}(\mathbf{x}) - \int \hat{p}(\mathbf{x}) \log\left(q(\mathbf{x}_F)q(\mathbf{x}_r|\mathbf{x}_F) \prod_{i \in \mathcal{V} \setminus F \setminus r} q(\mathbf{x}_i|\mathbf{x}_F, \mathbf{x}_{\pi(i)})\right) \mathrm{d}\mathbf{x} \\
&= -H_{\hat{p}}(\mathbf{x}) - \int \hat{p}(\mathbf{x}_F) \log q(\mathbf{x}_F) \mathrm{d}\mathbf{x}_F - \int \hat{p}(\mathbf{x}_F, \mathbf{x}_r) \log q(\mathbf{x}_r|\mathbf{x}_F) \mathrm{d}\mathbf{x}_F \mathrm{d}\mathbf{x}_r \\
&\quad - \sum_{i \in \mathcal{V} \setminus F \setminus r} \int \hat{p}(\mathbf{x}_F, \mathbf{x}_{\pi(i)}, \mathbf{x}_i) \log q(\mathbf{x}_i|\mathbf{x}_F, \mathbf{x}_{\pi(i)}) \mathrm{d}\mathbf{x}_F \mathrm{d}\mathbf{x}_{\pi(i)} \mathrm{d}\mathbf{x}_i \\
&\overset{(b)}{=} -H_{\hat{p}}(\mathbf{x}) + H_{\hat{p}}(\mathbf{x}_F) + D(\hat{p}_F||q_F) + H_{\hat{p}}(\mathbf{x}_r|\mathbf{x}_F) + D(\hat{p}_{r|F}||q_{r|F}|\hat{p}_F) \\
&\quad + \sum_{i \in V \setminus F \setminus r} H_{\hat{p}}(\mathbf{x}_i|\mathbf{x}_{F,\pi(i)}) + D(\hat{p}_{i|F,r}||q_{i|F,r}|\hat{p}_{F,r}) \\
&\overset{(c)}{\geq} -H_{\hat{p}}(\mathbf{x}) + H_{\hat{p}}(\mathbf{x}_F) + H_{\hat{p}}(\mathbf{x}_r|\mathbf{x}_F) + \sum_{i \in V \setminus F \setminus r} H_{\hat{p}}(\mathbf{x}_i|\mathbf{x}_{F,\pi(i)}), \quad (12)
\end{aligned}
$$

where (a) is obtained by using Factorization 1 in Lemma 2 with an arbitrary root node $r$; (b) can be directly verified using the definition of the information quantities, and the equality in (c) is satisfied when $q_F = \hat{p}_F$, $q_{r|F} = \hat{p}_{r|F}$, and $q_{i|F,\pi(i)} = \hat{p}_{i|F,\pi(i)}$, $\forall i \in T \setminus r$, or equivalently when

$$
\begin{aligned}
q_F &= \hat{p}_F \\
q_{F,i,j} &= \hat{p}_{F,i,j}, \forall (i,j) \in \mathcal{E}_{\mathcal{T}}. \quad (13)
\end{aligned}
$$

Next, we derive another expression of (12). By substituting (13) into Factorization s of Lemma 2, we have

$$q^*(\mathbf{x}) = \hat{p}(\mathbf{x}_F) \prod_{i \in T} \hat{p}(x_i|\mathbf{x}_F) \prod_{(i,j) \in \mathcal{E}_{\mathcal{T}}} \frac{\hat{p}(\mathbf{x}_i, \mathbf{x}_j|\mathbf{x}_F)}{\hat{p}(\mathbf{x}_i|\mathbf{x}_F)\hat{p}(\mathbf{x}_j|\mathbf{x}_F)}.$$

Hence,

$$\min_{q \in \mathcal{Q}_{F,\mathcal{T}}} D(\hat{p}||q) = D(\hat{p}||q^*)$$

$$- H_{\hat{p}}(\mathbf{x}) + H_{\hat{p}}(\mathbf{x}_F) + \sum_{i \in V \setminus F} H_{\hat{p}}(\mathbf{x}_i|\mathbf{x}_F) \tag{14}$$

$$+ \sum_{(i,j) \in \mathcal{E}_{\mathcal{T}}} \int \hat{p}_{F,i,j}(\mathbf{x}_F, \mathbf{x}_i, \mathbf{x}_j) \log \frac{\hat{p}(\mathbf{x}_i, \mathbf{x}_j|\mathbf{x}_F)}{\hat{p}(\mathbf{x}_i|\mathbf{x}_F)\hat{p}(\mathbf{x}_j|\mathbf{x}_F)} \mathrm{d}\mathbf{x}_F \mathrm{d}\mathbf{x}_i \mathrm{d}\mathbf{x}_j \tag{15}$$

$$= H_{\hat{p}}(\mathbf{x}) + H_{\hat{p}}(\mathbf{x}_F) + \sum_{i \in V \setminus F} H_{\hat{p}}(\mathbf{x}_i|\mathbf{x}_F) \tag{16}$$

$$- \sum_{(i,j) \in \mathcal{E}_{\mathcal{T}}} \int \hat{p}_{F,i,j}(\mathbf{x}_F, \mathbf{x}_i, \mathbf{x}_j) \log \frac{\hat{p}(\mathbf{x}_i|\mathbf{x}_F)\hat{p}(\mathbf{x}_j|\mathbf{x}_F)}{\hat{p}(\mathbf{x}_i, \mathbf{x}_j|\mathbf{x}_F)} \mathrm{d}\mathbf{x}_F \mathrm{d}\mathbf{x}_i \mathrm{d}\mathbf{x}_j \tag{17}$$

$$= -H_{\hat{p}}(\mathbf{x}) + H(\hat{p}_F) + \sum_{i \in \mathcal{V} \setminus F} H(\hat{p}_{i|F}|\mathbf{x}_F) - \sum_{(i,j) \in \mathcal{E}_{\mathcal{T}}} I_{\hat{p}}(\mathbf{x}_i; \mathbf{x}_j|\mathbf{x}_F). \tag{18}$$

We have thus proved Lemma 3.

$\square$

The following Lemma 4 gives a closed-form expression for the K-L divergence between two Gaussians. It can be verified by calculus and the proof is omitted.

**Lemma 4.** *For two $n$-dimensional Gaussian distributions $\hat{p}(\mathbf{x}) = \mathcal{N}(\mathbf{x}; \hat{\boldsymbol{\mu}}, \hat{\Sigma})$ and $q(\mathbf{x}) = \mathcal{N}(\mathbf{x}; \boldsymbol{\mu}, \Sigma)$, we have*

$$D(\hat{p}||q) = \frac{1}{2}\left( Tr\left(\Sigma^{-1}\hat{\Sigma}\right) + (\boldsymbol{\mu} - \hat{\boldsymbol{\mu}})^T \Sigma^{-1}(\boldsymbol{\mu} - \hat{\boldsymbol{\mu}}) - n \ln \det\left(\Sigma^{-1}\hat{\Sigma}\right) \right). \tag{19}$$

An immediate implication of Lemma 4 is that when learning GGMs we always have that $\boldsymbol{\mu}_{\mathrm{ML}} = \hat{\boldsymbol{\mu}}$ if there is no constraint on the mean in.

**Lemma 5.** *If a symmetric positive definite matrix $\Sigma$ is given and we know that its inverse $J = \Sigma^{-1}$ is sparse with respect to a tree $\mathcal{T} = (\mathcal{V}, \mathcal{E})$, then the non-zero entries of $J$ can be computed using (20) in time $\mathcal{O}(n)$.*

$$J_{ij} = \begin{cases} (1 - deg(i))\Sigma_{ii}^{-1} + \sum_{j \in \mathcal{N}(i)} \left(\Sigma_{ii} - \Sigma_{ij}\Sigma_{jj}^{-1}\Sigma_{ji}\right)^{-1} & i = j \in \mathcal{V} \\ \frac{\Sigma_{ij}}{\Sigma_{ij}^2 - \Sigma_{ii}\Sigma_{jj}} & (i,\,j) \in \mathcal{E} \\ 0 & otherwise, \end{cases} \tag{20}$$

*where $\mathcal{N}(i)$ is the set of neighbors of node $i$ in $\mathcal{T}$; $deg(i)$ is the degree of $i$ in $\mathcal{T}$.*

*Proof.* Since $\Sigma \succ 0$, we can construct a Gaussian distribution $p(\mathbf{x})$ with zero mean and covariance matrix $\Sigma$. The distribution is tree-structured because $J = \Sigma^{-1}$ has tree structure $\mathcal{T}$. Hence, we have the following factorization.

$$p(\mathbf{x}) = \prod_{i \in \mathcal{V}} p(x_i) \prod_{(i,j) \in \mathcal{E}} \frac{p(x_i, x_j)}{p(x_i)p(x_j)},$$

where

$$p(\mathbf{x}) = \frac{1}{(2\pi)^{\frac{n}{2}} \left(\det J\right)^{-\frac{1}{2}}} \exp\{-\frac{1}{2}\mathbf{x}^T J \mathbf{x}\}$$

$$p(x_i) = \frac{1}{(2\pi)^{\frac{1}{2}} P_{ii}^{\frac{1}{2}}} \exp\{-\frac{1}{2}\mathbf{x}^T \Sigma_{ii}^{-1}\mathbf{x}\}$$

$$p(x_i, x_j) = \frac{1}{2\pi \left(\det \begin{bmatrix} \Sigma_{ii} & \Sigma_{ij} \\ \Sigma_{ji} & \Sigma_{jj} \end{bmatrix}\right)^{\frac{1}{2}}} \exp\{-\frac{1}{2}\mathbf{x}^T \begin{bmatrix} \Sigma_{ii} & \Sigma_{ij} \\ \Sigma_{ji} & \Sigma_{jj} \end{bmatrix}^{-1}\mathbf{x}\}.$$

By matching the quadratic coefficient in the exponents, we have that

$$J_{ii} = \Sigma_{ii}^{-1} + \sum_{j \in \mathcal{N}(i)} \left( \left( \begin{bmatrix} \Sigma_{ii} & \Sigma_{ji} \\ \Sigma_{ij} & \Sigma_{jj} \end{bmatrix}^{-1} \right)_{11} - \Sigma_{ii}^{-1} \right)$$

$$= (1 - \deg(i))\Sigma_{ii}^{-1} + \sum_{j \in \mathcal{N}(i)} \left( \Sigma_{ii} - \Sigma_{ij}\Sigma_{jj}^{-1}\Sigma_{ji} \right)^{-1}$$

and for $(i, j) \in \mathcal{E}$,

$$J_{ij} = \left( \begin{bmatrix} \Sigma_{ii} & \Sigma_{ij} \\ \Sigma_{ji} & \Sigma_{jj} \end{bmatrix}^{-1} \right)_{12}$$

$$= \frac{\Sigma_{ij}}{\Sigma_{ij}^2 - \Sigma_{ii}\Sigma_{jj}}$$

The complexity of computing each $J_{ij}$, $(i, j) \in \mathcal{E}$ is $O(1)$ and the complexity of computing each $J_{ii}$ is $O(\deg i)$. Since $\Sigma_{i \in \mathcal{V}} \deg(i)$ equals twice the number of edges, which is $\mathcal{O}(n)$, the total complexity is $\mathcal{O}(n)$.

$\square$

**Lemma 6.** *(The matrix inversion lemmas)*

*If* $\begin{bmatrix} A & B \\ C & D \end{bmatrix}$ *is invertible, we have*

$$\begin{bmatrix} A & B \\ C & D \end{bmatrix}^{-1} = \begin{bmatrix} (A - BD^{-1}C)^{-1} & -(A - BD^{-1}C)^{-1}BD^{-1} \\ -D^{-1}C(A - BD^{-1}C)^{-1} & D^{-1} + D^{-1}C(A - BD^{-1}C)^{-1}BD^{-1} \end{bmatrix} \quad (21)$$

*or*

$$\begin{bmatrix} A & B \\ C & D \end{bmatrix}^{-1} = \begin{bmatrix} A^{-1} + A^{-1}B(D - CA^{-1}B)^{-1}CA^{-1} & -A^{-1}B(D - CA^{-1}B)^{-1} \\ -(D - CA^{-1}B)^{-1}CA^{-1} & (D - CA^{-1}B)^{-1} \end{bmatrix} \quad (22)$$

*and*

$$\left( A - BD^{-1}C \right)^{-1} = A^{-1} + A^{-1}B(D - CA^{-1}B)^{-1}CA^{-1}. \quad (23)$$

The proof of Lemma 6 can be found in standard matrix analysis books.

### B.3   Proof of Proposition 1

*Proof.* For a fixed FVS $F$, the LHS of (11) is only a function of the spanning tree among the non-feedback nodes. Hence, the optimal set of edges among the non-feedback nodes can be obtained by finding the maximum spanning tree of the subgraph induced by $T$ with $I_{\hat{p}}(\mathbf{x}_i; \mathbf{x}_j|\mathbf{x}_F) \geq 0$ being the edge weight between $i$ and $j$. [4]

For Gaussian distributions, the covariance matrix of the distribution $\hat{p}(\mathbf{x}_T|\mathbf{x}_F)$ depends only on the set $F$ but is invariant to the value of $\mathbf{x}_F$. Hence, finding the optimal edge set of the tree part is equivalent to running the Chow-Liu algorithm with the input being the covariance matrix of $\hat{p}_{T|F}(\mathbf{x}_T|\mathbf{x}_F)$, which is simply $\hat{\Sigma}_{T|F} = \hat{\Sigma}_T - \hat{\Sigma}_M \hat{\Sigma}_F^{-1} \hat{\Sigma}_M^T$. Let $\mathcal{E}_{\mathrm{CL}} = \mathrm{CL}_{\mathcal{E}}(\hat{\Sigma}_{T|F})$ and $\Sigma_{\mathrm{CL}} = \mathrm{CL}(\hat{\Sigma}_{T|F})$. Denote the optimal covariance matrix as $\Sigma_{\mathrm{ML}} = \begin{bmatrix} \Sigma_F^{\mathrm{ML}} & (\Sigma_M^{\mathrm{ML}})^T \\ \Sigma_M^{\mathrm{ML}} & \Sigma_T^{\mathrm{ML}} \end{bmatrix}$. According to (13), we must have $\Sigma_F^{\mathrm{ML}} = \hat{\Sigma}_F$ and $\Sigma_M^{\mathrm{ML}} = \hat{\Sigma}_M$. From (13) the corresponding conditional covariance matrix $\Sigma_{T|F}^{\mathrm{ML}}$ of $\Sigma_{\mathrm{ML}}$ must equal $\Sigma_{\mathrm{CL}}$. Hence, we have $\Sigma_{T|F}^{\mathrm{ML}} = \Sigma_T^{\mathrm{ML}} - \Sigma_M^{\mathrm{ML}} (\Sigma_F^{\mathrm{ML}})^{-1} (\Sigma_M^{\mathrm{ML}})^T = \Sigma_{\mathrm{CL}}$. Therefore, we can obtain $\Sigma_T^{\mathrm{ML}} = \mathrm{CL}(\hat{\Sigma}_{T|F}) + \hat{\Sigma}_M \hat{\Sigma}_F^{-1} \hat{\Sigma}_M^T$. We also have that $\mathcal{E}_{\mathrm{ML}} = \mathcal{E}_{\mathrm{CL}}$ since $\mathcal{E}_{\mathrm{ML}}$ is defined to be the set of edges among the feedback nodes.

Now we analyze the complexity of Algorithm 1. The matrix $\hat{\Sigma}_{T|F}$ is computed with complexity $\mathcal{O}(kn^2)$. Computing the maximum weight spanning tree algorithm has complexity $\mathcal{O}(n^2 \log n)$ using Kruskal's algorithm (the amortized complexity can be further reduced, but it is not the focus of this paper). Other operations have complexity $\mathcal{O}(n^2)$. Hence, the total complexity of Algorithm 1 is $\mathcal{O}(kn^2 + n^2 \log n)$.

Next we proceed to prove that we can compute all the non-zero entries of $J_{\mathrm{ML}} = (\Sigma_{\mathrm{ML}})^{-1}$ in time $\mathcal{O}(k^2 n)$.

Let $J_{\mathrm{ML}} = \begin{bmatrix} J_F^{\mathrm{ML}} & (J_M^{\mathrm{ML}})^T \\ J_M^{\mathrm{ML}} & J_T^{\mathrm{ML}} \end{bmatrix}$. We have that $J_T^{\mathrm{ML}} = \left(\mathrm{CL}(\hat{\Sigma}_{T|F})\right)^{-1}$ has tree structure with $\mathcal{T}$. Therefore, the non-zero entries of $J_T^{\mathrm{ML}}$ can be computed with complexity $\mathcal{O}(n-k)$ from Lemma 5.

From (22) we have

$$J_M^{\mathrm{ML}} = -J_T^{\mathrm{ML}} \Sigma_M^{\mathrm{ML}} (\Sigma_F^{\mathrm{ML}})^{-1}, \tag{24}$$

which can be computed with complexity $\mathcal{O}(k^2 n)$ by matrix multiplication in the regular order. Note that $J_T^{\mathrm{ML}} \Sigma_M^{\mathrm{ML}}$ is computed in $\mathcal{O}(kn)$ since $J_T^{\mathrm{ML}}$ only has $\mathcal{O}(n)$ non-zero entries.

From (22) we have

$$J_F^{\mathrm{ML}} = (\Sigma_F^{\mathrm{ML}})^{-1} \left( I + \left((\Sigma_M^{\mathrm{ML}})^T J_T^{\mathrm{ML}}\right) \left(\Sigma_M^{\mathrm{ML}} (\Sigma_F^{\mathrm{ML}})\right) \right),$$

which has complexity $\mathcal{O}(k^2 n)$ following the order specified by the parentheses. Note that $\left(P_M^{\mathrm{ML}}\right)^T J_T^{\mathrm{ML}}$ is computed in $\mathcal{O}(kn)$ because $J_T^{\mathrm{ML}}$ only has $\mathcal{O}(n)$ non-zero entries. Hence, we need extra complexity of $\mathcal{O}(k^2 n)$ to compute all the non-zero entries of $J_{\mathrm{ML}}$.

We have therefore completed the proof for Proposition 1.

$\square$

For easy reference, we summarize the procedure to compute $J_{\mathrm{ML}}$ in Algorithm 6.

---

**Algorithm 6** Compute $J_{\mathrm{ML}} = (\Sigma_{\mathrm{ML}})^{-1}$ after running Algorithm 1

1. Compute $J_T^{\mathrm{ML}}$ using (20)

2. Compute $J_M^{\mathrm{ML}} = -J_T^{\mathrm{ML}} \Sigma_M^{\mathrm{ML}} \Sigma_F^{-1}$ using sparse matrix multiplication

3. Compute $(\Sigma_F^{\mathrm{ML}})^{-1} \left( I + \left((\Sigma_M^{\mathrm{ML}})^T J_T^{\mathrm{ML}}\right) \left(\Sigma_M^{\mathrm{ML}} (\Sigma_F^{\mathrm{ML}})\right) \right)$ following the order specified by the parentheses using sparse matrix multiplication.

---

## C  Proof of Proposition 2

In this section, we first prove a more general result stated in Lemma 7.

**Lemma 7.** *In Algorithm 7, if Step 2(a) and Step 2(b) can be computed exactly, then we have that $D(\hat{p}(\mathbf{x}_T)||q^{(t+1)}(\mathbf{x}_T)) \leq D(\hat{p}(\mathbf{x}_T)||q^{(t)}(\mathbf{x}_T))$, where the equality is satisfied if and only if $\hat{p}^{(t)}(\mathbf{x}_F, \mathbf{x}_T) = \hat{p}^{(t+1)}(\mathbf{x}_F, \mathbf{x}_T)$.*

---

**Algorithm 7** Alternating Projection

1. Propose an initial distribution $q^{(0)}(\mathbf{x}_F, \mathbf{x}_T) \in \mathcal{Q}_F$

2. Alternating projections:

   (a) **P1:** Project to the empirical distribution:

   $$\hat{p}^{(t)}(\mathbf{x}_F, \mathbf{x}_T) = \hat{p}(\mathbf{x}_T) q^{(t)}(\mathbf{x}_F|\mathbf{x}_T)$$

   (b) **P2:** Project to the best fitting structure on all variables:

   $$q^{(t+1)}(\mathbf{x}_F, \mathbf{x}_T) = \arg\min_{q(\mathbf{x}_F, \mathbf{x}_T) \in \mathcal{Q}_F} D(\hat{p}^{(t)}(\mathbf{x}_F, \mathbf{x}_T)||q(\mathbf{x}_F, \mathbf{x}_T))$$

   .

---

*Proof.* For any $t$,

$$D(\hat{p}^{(t)}(\mathbf{x}_T, \mathbf{x}_F)||q^{(t)}(\mathbf{x}_F, \mathbf{x}_T))$$

$$= \int_{\mathbf{x}_T, \mathbf{x}_F} \hat{p}(\mathbf{x}_T) q^{(t)}(\mathbf{x}_F|\mathbf{x}_T) \log \frac{\hat{p}(\mathbf{x}_T) q^{(t)}(\mathbf{x}_F|\mathbf{x}_T)}{q^{(t)}(\mathbf{x}_F, \mathbf{x}_T)}$$

$$= \int_{\mathbf{x}_T, \mathbf{x}_F} \hat{p}(\mathbf{x}_T) q^{(t)}(\mathbf{x}_F|\mathbf{x}_T) \log \frac{\hat{p}(\mathbf{x}_T)}{q^{(t)}(\mathbf{x}_T)}$$

$$= \int_{\mathbf{x}_T} \hat{p}(\mathbf{x}_T) \log \frac{\hat{p}(\mathbf{x}_T)}{q^{(t)}(\mathbf{x}_T)}$$

$$= D(\hat{p}^{(t)}(\mathbf{x}_T)||q^{(t)}(\mathbf{x}_T)) \tag{25}$$

By the definition of $q^{(t+1)}$ in step (b), we have

$$D(\hat{p}(\mathbf{x}_T, \mathbf{x}_F)||q^{(t+1)}(\mathbf{x}_F, \mathbf{x}_T)) \leq D(\hat{p}^{(t)}(\mathbf{x}_T, \mathbf{x}_F)||q^{(t)}(\mathbf{x}_F, \mathbf{x}_T)). \tag{26}$$

Therefore,

$$D(\hat{p}(\mathbf{x}_T)||q^{(t)}(\mathbf{x}_T))$$

$$\stackrel{(a)}{=} D(\hat{p}^{(t)}(\mathbf{x}_T, \mathbf{x}_F)||q^{(t)}(\mathbf{x}_F, \mathbf{x}_T)) \tag{27}$$

$$\stackrel{(b)}{\geq} D(\hat{p}^{(t)}(\mathbf{x}_T, \mathbf{x}_F)||q^{(t+1)}(\mathbf{x}_F, \mathbf{x}_T)) \tag{28}$$

$$= \int_{\mathbf{x}_T, \mathbf{x}_F} \hat{p}(\mathbf{x}_T) q^{(t)}(\mathbf{x}_F|\mathbf{x}_T) \log \frac{\hat{p}(\mathbf{x}_T) q^{(t)}(\mathbf{x}_F|\mathbf{x}_T)}{q^{(t+1)}(\mathbf{x}_F, \mathbf{x}_T)}$$

$$= \int_{\mathbf{x}_T, \mathbf{x}_F} \hat{p}(\mathbf{x}_T) q^{(t)}(\mathbf{x}_F|\mathbf{x}_T) \log \frac{\hat{p}(\mathbf{x}_T)}{q^{(t+1)}(\mathbf{x}_T)} + \int_{\mathbf{x}_T, \mathbf{x}_F} \hat{p}(\mathbf{x}_T) q^{(t)}(\mathbf{x}_F|\mathbf{x}_T) \log \frac{q^{(t)}(\mathbf{x}_F|\mathbf{x}_T)}{q^{(t+1)}(\mathbf{x}_F|\mathbf{x}_T)}$$

$$= \int_{\mathbf{x}_T} \hat{p}(\mathbf{x}_T) \log \frac{\hat{p}(\mathbf{x}_T)}{q^{(t+1)}(\mathbf{x}_T)} + \int_{\mathbf{x}_T, \mathbf{x}_F} \hat{p}(\mathbf{x}_T) q^{(t)}(\mathbf{x}_F|\mathbf{x}_T) \log \frac{q^{(t)}(\mathbf{x}_F|\mathbf{x}_T) \hat{p}(\mathbf{x}_T)}{q^{(t+1)}(\mathbf{x}_F|\mathbf{x}_T) \hat{p}(\mathbf{x}_T)} \tag{29}$$

$$= D(\hat{p}(\mathbf{x}_T)||q^{(t+1)}(\mathbf{x}_T)) + \int_{\mathbf{x}_T, \mathbf{x}_F} \hat{p}^{(t)}(\mathbf{x}_F, \mathbf{x}_T) \log \frac{\hat{p}^{(t)}(\mathbf{x}_F, \mathbf{x}_T)}{\hat{p}^{(t+1)}(\mathbf{x}_F, \mathbf{x}_T)}$$

$$= D(\hat{p}(\mathbf{x}_T)||q^{(t+1)}(\mathbf{x}_T)) + D(\hat{p}^{(t)}(\mathbf{x}_F, \mathbf{x}_T)||\hat{p}^{(t+1)}(\mathbf{x}_F, \mathbf{x}_T))$$

$$\stackrel{(c)}{\geq} D(\hat{p}(\mathbf{x}_T)||q^{(t+1)}(\mathbf{x}_T)), \tag{30}$$

where (a) is due to (25), (b) is due to (26), and (c) is due to that $D(\hat{p}^{(t)}(\mathbf{x}_F, \mathbf{x}_T)||\hat{p}^{(t+1)}(\mathbf{x}_F, \mathbf{x}_T)) \geq 0$. Therefore, we always have $D(\hat{p}(\mathbf{x}_T)||q^{(t)}) \geq D(\hat{p}(\mathbf{x}_T)||q^{(t+1)})$. A necessary condition for the objective function to remain the same is that $D(\hat{p}^{(t)}(\mathbf{x}_F, \mathbf{x}_T)||\hat{p}^{(t+1)}(\mathbf{x}_F, \mathbf{x}_T)) = 0$, which implies that $\hat{p}^{(t)}(\mathbf{x}_F, \mathbf{x}_T) = \hat{p}^{(t+1)}(\mathbf{x}_F, \mathbf{x}_F)$. Hence, it further implies that $q^{(t)}(\mathbf{x}_F, \mathbf{x}_T) = q^{(t+1)}(\mathbf{x}_F, \mathbf{x}_T)$ under non-degenerate cases. Therefore, $\hat{p}^{(t)}(\mathbf{x}_F, \mathbf{x}_T) = \hat{p}^{(t+1)}(\mathbf{x}_F, \mathbf{x}_F)$ is a necessary and sufficient condition for the objective function to remain the same. This completes the proof for Lemma 7.

$\square$

Now we proceed to the proof for Proposition 2.

*Proof.* Use the same notation as in the latent Chow-Liu algorithm (Algorithm 3). Let $\hat{p}(\mathbf{x}_T) = \mathcal{N}(\mathbf{0}, \hat{\Sigma}_T), p^{(t)}(\mathbf{x}_F, \mathbf{x}_T) = \mathcal{N}(\mathbf{0}, \Sigma^{(t)})$. Then

$$\hat{p}(\mathbf{x}_T) = \frac{1}{\sqrt{\det\left(2\pi\hat{\Sigma}_T\right)}} \exp\{-\frac{1}{2}\mathbf{x}_T^T\hat{\Sigma}_T^{-1}\mathbf{x}_T\}$$

$$p^{(t)}(\mathbf{x}_F|\mathbf{x}_T) = \frac{1}{\sqrt{\det\left(2\pi\left(J_F^{(t)}\right)^{-1}\right)}} \exp\{-\frac{1}{2}\left(\mathbf{x}_F - \left(J_F^{(t)}\right)^{-1}J_M^{(t)}\mathbf{x}_T\right)^T J_F^{(t)} \left(\mathbf{x}_F - \left(J_F^{(t)}\right)^{-1}J_M^{(t)}\mathbf{x}_T\right)^T\}$$

Hence, following Algorithm 7, we have

$$\hat{p}^{(t)}(\mathbf{x}_F, \mathbf{x}_T) = \hat{p}(\mathbf{x}_T)q^{(t)}(\mathbf{x}_F|\mathbf{x}_T)$$

$$\propto \exp\{-\frac{1}{2}\left[\begin{array}{c} \mathbf{x}_F \\ \mathbf{x}_T \end{array}\right]^T \left[\begin{array}{cc} J_F^{(t)} & \left(J_M^{(t)}\right)^T \\ J_M^{(t)} & \hat{\Sigma}_T^{-1} + J_M^{(t)}(J_F^{(t)})^{-1}(J_M^{(t)})^T \end{array}\right] \left[\begin{array}{c} \mathbf{x}_F \\ \mathbf{x}_T \end{array}\right]\},$$

which gives the same expression as in **P1** of Algorithm 3. The next projection

$$q^{(t+1)}(\mathbf{x}_F, \mathbf{x}_T) = \min_{q(\mathbf{x}_F, \mathbf{x}_T) \in \mathcal{Q}_F} D(\hat{p}^{(t)}(\mathbf{x}_F, \mathbf{x}_T)||q(\mathbf{x}_F, \mathbf{x}_T))$$

has same form as M-L learning problem in Section 4.1.1, and therefore can be computed exactly using the conditioned Chow-Liu algorithm (Algorithm 1). By Lemma 7, we have thus proved the first part of Proposition 2. The second part about the complexity of an accelerated version is proved in Section D.

$\square$

# D   The Accelerated Latent Chow-Liu Algorithm

In this section, we describe the accelerated latent Chow-Liu algorithm (Algorithm 8), which gives exactly the same result as the latent Chow-Liu algorithm 3, but has a lower complexity of $\mathcal{O}(kn^2 + n^2 \log n)$ per iteration. The main complexity reduction is due to the use of Algorithm 6.

We will use the following lemma in the proof of Algorithm 8.

Now we proceed to prove the correctness of the accelerated Chow-Liu algorithm and obtain its complexity.

*Proof.* In **P1** of the latent Chow-Liu algorithm (Algorithm 3) we have

$$\hat{J}^{(t)} = \left[\begin{array}{cc} J_F^{(t)} & (J_M^{(t)})^T \\ J_M^{(t)} & \left(\hat{\Sigma}_T\right)^{-1} + J_M^{(t)}(J_F^{(t)})^{-1}(J_M^{(t)})^T \end{array}\right].$$

Without explicitly computing $\hat{J}^{(t)}$, we can directly compute $\hat{\Sigma}^{(t)} = \left(\hat{J}^{(t)}\right)^{-1}$ as follows.

Let $A = J_F^{(t)}$, $B = (J_M^{(t)})^T$, $C = \hat{J}_M^{(t)}$, and $D = \left(\hat{\Sigma}_T\right)^{-1} + J_M^{(t)}(J_F^{(t)})^{-1}(J_M^{(t)})^T)$. From (22) we have

$$\hat{\Sigma}_F^{(t)} = \left(J_F^{(t)}\right)^{-1} + \left(J_F^{(t)}\right)^{-1}\left(J_M^{(t)}\right)^T (D - CA^{-1}B)^{-1}\hat{J}_M^{(t)}\left(J_F^{(t)}\right)^{-1}$$

and

$$\hat{\Sigma}_T^{(t)} = (D - CA^{-1}B)^{-1} = \hat{\Sigma}_T. \tag{31}$$

$$\hat{\Sigma}_F^{(t)} = \left(J_F^{(t)}\right)^{-1} + \left(J_F^{(t)}\right)^{-1}\left(J_M^{(t)}\right)^T \hat{\Sigma}_T \hat{J}_M^{(t)}\left(J_F^{(t)}\right)^{-1}. \tag{32}$$

Also from (22), we have that

$$\hat{\Sigma}_M^{(t)} = -\hat{\Sigma}_T J_M^{(t)}\left(J_F^{(t)}\right)^{-1}. \tag{33}$$

It can be checked that the matrix multiplications of (31), (32), and (33) have complexity $\mathcal{O}(kn^2)$.

**P2** in Algorithm 3 can be computed with complexity $\mathcal{O}(n^2 k + n^2 \log n)$ from Proposition 1. Therefore, the complexity of this accelerated version (summarized in Algorithm 8) is $\mathcal{O}(n^2 k + n^2 \log n)$ per iteration. We have thus completed the proof for Proposition 2.

---

**Algorithm 8** The accelerated Chow-Liu algorithm

---

**Input:** the empirical covariance matrix $\hat{\Sigma}_T$

**Output:** information matrix $J = \begin{bmatrix} J_{\mathcal{F}} & J_M^T \\ J_M & J_{\mathcal{T}} \end{bmatrix}$.

1. Initialization: $J^{(0)} = \begin{bmatrix} J_F^{(0)} & \left(J_M^{(0)}\right)^T \\ J_M^{(0)} & J_T^{(0)} \end{bmatrix}$.

2. Repeat

   (a) **AP1:** Compute

   $$\hat{\Sigma}_F^{(t)} = \left(J_F^{(t)}\right)^{-1} + \left(Y^{(t)}\right)^T \hat{\Sigma}_T Y^{(t)}$$
   $$\hat{\Sigma}_T^{(t)} = \hat{\Sigma}_T$$
   $$\hat{\Sigma}_M^{(t)} = -\hat{\Sigma}_T Y^{(t)},$$

   where $Y^{(t)} = \hat{J}_M^{(t)}\left(J_F^{(t)}\right)^{-1}$

   Let $\hat{\Sigma}^{(t)} = \begin{bmatrix} \hat{\Sigma}_F^{(t)} & \left(\hat{\Sigma}_M^{(t)}\right)^T \\ \hat{\Sigma}_M^{(t)} & \hat{\Sigma}_T \end{bmatrix}$.

   (b) **AP2:** Compute $\Sigma^{(t+1)}$ and $J^{(t+1)} = \left(\Sigma^{(t+1)}\right)^{-1}$ from $\hat{\Sigma}^{(t)}$ using Algorithm 1 and Algorithm 6:

   $$J^{(t+1)} = \begin{bmatrix} J_F^{(t+1)} & \left(J_M^{(t+1)}\right)^T \\ J_M^{(t+1)} & J_T^{(t+1)} \end{bmatrix}$$

   $$\Sigma^{(t+1)} = \begin{bmatrix} \Sigma_F^{(t+1)} & \left(\Sigma_M^{(t+1)}\right)^T \\ \Sigma_M^{(t+1)} & \Sigma_T^{(t+1)} \end{bmatrix}$$

---

$\square$

## Footnotes

[4] In fact, we have given an algorithm to learn general models (not only GGMs, but also other models, e.g., discrete ones) defined on graphs with a given FVS $F$. However, we do not explore the general setting in this paper.