[Reviews · NeurIPS 2013]

Submitted by Assigned_Reviewer_1

'Learning Gaussian Graphical Models with Observed or Latent FVSs' addresses the problem of learning (i.e. discovery of the graph structure) a GGM which can be decomposed into a small feedback vertex set and a tree. The motivation is that exact inference under these models can be done quickly, and so in the case where one needs near-linear inference (which is prohibited in general for sparse GGMs) it is desirable to have this form. The results address three cases: (4.1.1) all nodes are observed with known FVS, (4.1.2) all nodes are observed with unknown FVS, (4.2) and the FVS is latent. In (4.1.1) they provide a simple extension of the Chow-Liu algorithm, which applies CL to the empirical Schur complement. In (4.1.2) they make the observation that one can exhaustively run the previous algorithm for all k-sets selecting the one that maximizes the likelihood and then provide a greedy algorithm. In (4.2) they provide an EM-type algorithm that uses the conditional distribution from the previous iteration to find an empirical joint distribution then finding the best structure from this with CL. There is a nice interpretation using low rank adjustments in each step. They show that each iteration decreases the KL. There is a very nice experimental section, which applies this to flight traffic data.

The paper is high quality, as it motivates the problem well, describes the landscape of learning GGMs, and comprehensively provides an algorithm for each situation. The paper is clear as it is well written and well organized. The paper seems original, as it is a unique extension of the Chow-Liu algorithm. The significance of this paper is somewhat suspect (although I admit that I am not an expert in this field). While it is the case that exact inference is more difficult for non-tree graphs, it is a bit difficult to imagine real-world GGMs that can be decomposed into the FVS and a tree. And if we don't believe that we are providing an accurate graph and only a convenient approximation, then approximate inference should be as good as we can hope for. If this is the case, then we can employ loopy BP, or even linear solvers. Prop 1 is really just the fact that CL works on the Schur complement too, and Prop 2 doesn't tell us that much. Again, I may have overlooked some fundamental insight into how important this extension is, so please argue with me on this point. With that said, I think that the insights into using CL with the FVS merits publication, as there is precedent for doing inference with the FVS.

typos:
141: proved in the
265: the sentence 'We have no search of sets...' is awkward
306: overlaping lines
313: D_KL(N(0,...
Summary: This paper is well written, the work is well motivated and the algorithms are justified. While it is not clear how well FVSs can be used to model real-world GGMs, it is a natural extension of Chow-Liu algorithm, and so this paper deserves publication.

Submitted by Assigned_Reviewer_5

The abstract correctly summarises the paper: the focus is structure learning for GGMs with a limit on the number of feedback vertices (FVs). These FVs can be known/not known and observed or latent. Efficient inference within the learning provides an advantage (over a naive approach). Empirical results back up the claims for, and analyses of, the algorithms.

This is a great paper: with useful results and written with clarity. The structure of the paper is logical. I struggled to come up with criticisms, but failed. There's some missing capitalisation in the refs and a missing "in" on line 142, but that's it!

Throughout the paper I was wondering why there was no mention of regularisation/prior bias, but that was eventually mentioned at the end. I think it would be more natural to mention it earlier.
Summary: An excellent paper which provides useful learning algorithms for an important task. The structure and clarity of English is impressive.

Submitted by Assigned_Reviewer_7

The paper addresses the problem of learning a Gaussian graphical model with feedback vertex sets (FVSs),
where the graphical model has a tree structure given the variables in FVSs. The paper discusses
learning algorithms for various scenarios, including when the FVS is known or unknown (and needs to be learned)
or when the variables in the FVS are observed or unobserved.

The paper builds on the previous work on reducing the computational cost of inference
in a Gaussian graphical model by taking advantage of FVSs and discusses learning algorithms.
The paper provides a nice and quite complete discussion on the learning methods,
including various learning scenarios and time complexity.

When FVS is unknown, how should one choose the number of FVS? For example, in Figure 4, if one were to
further increase the size of FVS, would KL divergence further decrease? If the proposed approach tries
to find the smallest set for FVS, how can one ensure that the set is minimal?

When FVS is unknown, one needs to resort to a greedy approach. However, the experimental results provide
only an anectotal evidence based on a single dataset in Figure 4 that the greedy approach seems to work.
It does not provide any quantitative results on how accurately the true FVSs can be recovered.

In Figure 2, I wonder if the blocky structure in CLRG and NJ is because of the number of latent variables > 1,
whereas for the proposed method, FVS of size 1 was used.
Summary: Though the proposed approach may not be significantly novel, the paper provides
a complete discussion of the learning algorithms for Gaussian graphical models with FVS
that would be of interest to many people in the community. However, the experiments
are not complete and this section should be strengthend to provide evidence on
the accuracy of the models recovered by the proposed methods.
Author Feedback

Author rebuttal: We sincerely thank the reviewers for their careful read of the paper and for providing many constructive comments. We respond as follows.

Reviewer 1:

a) Much of the work on learning graphical models from data focuses on learning models with some desirable structure---e.g., sparsity, thin junction trees, and our FVS-based models, the latter two being motivated by constructing models that lead to efficient inference. A fair question is why our class of models is a good one. There are several reasons one can posit models: (a) they lead to efficient inference algorithms; (b) there are effective algorithms for learning them (note that ML estimation for thin junction trees is NP-hard); (c) they provide insight in terms of identifying variables and nodes that are “key” in terms of capturing statistical structures; and (d) they can provide accurate approximations to a rich class of phenomena. The first two of these are established by this paper; the third and fourth are important and we can address them through experimentation.

b) In particular, we intend to do a better job in the experiment section. We will provide results demonstrating how good our approximations are as a function of FVS size, which relates to the reviewer’s comment on approximating real-world GGMs. For very large problems the computational load of inference using direct linear solvers grows far more quickly than if we have a good approximate model that admits efficient inference. Also, we will include some results starting with a graphical model on a 2-D grid of a moderate size so that we can compute exact inference by brute force and also implement loopy BP as the reviewer suggests. We will also learn approximate models with small FVSs and then compare, as a function of FVS size, how accurate the models and the results are for the two inference algorithms (exact model+LBP versus approximateFVS-model+exact inference). We should say that there will be large classes of models in our favor, namely models that have long-distance correlations for which LBP diverges. To be fair, we will also show cases in which LBP does just fine.

c) The reviewer is right that the high level idea of the conditional CL algorithm is exactly CL on the Schur complement, and we agree that that provides useful insight and intuition. The technical difficulty lies in proving that the Schur complement is a sufficient statistic that characterizes the whole effect of the structure of the tree-part on the K-L divergence. This requires a bit of work as well as a special property of Gaussian distributions (The conditional mutual information depends only on the set conditioned on). For Prop 2, one common difficulty in applying the EM algorithm is that the E step is often intractable to compute and thus only approximate steps are used (in this case, the monotonicity is not satisfied). In our algorithm, we are able find a formulation that can convert the exact computation of the E step to exact learning in the case of observed FVS, which can be solved exactly by the conditional CL algorithm.

Reviewer 5:

d) We agree with the reviewer that it would be more natural to mention regularization/prior bias earlier. We will make this revision in the final version of the paper.

Reviewer 7:

e) We should have made it clear that in the experiments we either know the size of the FVS in advance or we can afford to test a few different sizes. We use this setting because the size of the FVS can take at most n different values (comparing with exponentially many different sets). Indeed in Figure 4, if one were to further increase the size of the FVS, the KL divergence would further decrease. However, when the size of the FVS increases from 3 to 4, the decrement in KL divergence would be rather small (comparing with from 0.6074 of size 2 to 0.0048 of size 3). This is related to the regularization/bias issue, and a systematic approach, which we propose but have not yet incorporated, is to include complexity into the FVS-order problem (e.g., as in AIC or BIC).

f) We agree with the reviewer that the experimental section can be much strengthened by including more quantitative results on the recovery rate of the true FVSs. In our previous experiments (which are not shown in the paper), the error rate seems to decay exponentially with the number of samples. In the final version, we will include such quantitative results and we will work on providing theoretical results of the recovery rate in our future work.

g) In addition, we will include experimental results that provide more empirical evidence of the claim that there is a rich class of processes that can be well-modeled by our models (see Paragraph (b) as well). We will show via experiments that by the time we get to log(n) we've captured all that needs to be captured. We contend that including at most a logarithmic number is sufficient for a large class of processes, and we will include experimental results to back that up.

h) The “blocky artifact” in latent tree models has been observed in many previous works (c.f. Figure 8 of [4] and “DTs: Dynamic Trees” by C. Williams and N. Adams) and occurs regardless of the number of latent variables. The reason is that two spatially adjacent variables may be far away in the underlying tree model if they happen to lie in different major branches. Because of the correlation decay (with respect to the distance on trees) in tree-structured models, the observed covariance matrix may have sharp transition (resulting in blocky artifacts) at these nodes. In our model, the blocky structure does not occur because the FVS captures long-distance correlation such that any two observed nodes have distance at most two in the latent graph. By adding some "long-distance" correlations through hidden nodes, we can capture much more global correlations.